# Modes of flagellar assembly in *Chlamydomonas reinhardtii* and *Trypanosoma brucei*

Johanna L Höög[1,2]*[†], Sylvain Lacomble[1], Eileen T O'Toole[2], Andreas Hoenger[2], J Richard McIntosh[2], Keith Gull[1]

[1]Sir William Dunn School of Pathology, University of Oxford, Oxford, United Kingdom; [2]The Boulder Laboratory for 3D Electron Microscopy of Cells, Department of MCD Biology, University of Colorado Boulder, Boulder, United States

**Abstract** Defects in flagella growth are related to a number of human diseases. Central to flagellar growth is the organization of microtubules that polymerize from basal bodies to form the axoneme, which consists of hundreds of proteins. Flagella exist in all eukaryotic phyla, but neither the mechanism by which flagella grow nor the conservation of this process in evolution are known. Here, we study how protein complexes assemble onto the growing axoneme tip using (cryo) electron tomography. In *Chlamydomonas reinhardtii* microtubules and associated proteins are added simultaneously. However, in *Trypanosoma brucei*, disorganized arrays of microtubules are arranged into the axoneme structure by the later addition of preformed protein complexes. Post assembly, the *T. brucei* transition zone alters structure and its association with the central pair loosens. We conclude that there are multiple ways to form a flagellum and that species-specific structural knowledge is critical before evaluating flagellar defects.

*For correspondence: hoog@mpi-cbg.de

†Present address: Max Planck Institute of Molecular Cell Biology and Genetics, Dresden, Germany

Competing interests: The authors declare that no competing interests exist.

## Introduction

Most vertebrate cells have a cilium or flagellum (the terms are used here interchangeable). It is a thin, microtubule-containing, membrane-covered extension on the surface of the cell, which may generate cell motility and/or act as a sensory and signaling organelle. Defects in cilia and flagella can cause severe human diseases for example skeletal deformations, polycystic kidney disease, infertility, situs invertus, blindness, obesity, and even cancer (*Escudier et al., 2009*; *Chandok, 2012*; *Huber and Cormier-Daire, 2012*; *Li et al., 2012*). A collective name for these conditions is the 'ciliopathies' (*Fliegauf et al., 2007*). Some ciliopathies, such as primary ciliary dyskinesia, are diagnosed by ultrastructural investigations of cilia that should be motile (*Chandok, 2012*; *Chilvers et al., 2003*; *Escudier et al., 2009*; *Huber and Cormier-Daire, 2012*; *Li et al., 2012*; *O'Toole et al., 2012*). However, the ultrastructural pathology of many ciliopathies remains unknown.

Each flagellum consists of ~1000 different proteins (*Pazour, 2005*; *Gherman et al., 2006*; *Fliegauf et al., 2007*; *Ishikawa et al., 2012*), many of which contribute to its microtubule-based core, called the axoneme. Axonemes originate inside the cell at basal bodies (*Figure 1A,B*). From the basal body, nine doublet microtubules (dMTs) extend into the next section along the flagellum called the transition zone. The transition zone ends at the basal plate, an electron-dense structure found in the region, where the two central pair microtubules (CPs) are nucleated and the canonical 9+2 axoneme arrangement starts. The dMTs consist of a complete A-tubule containing 13 protofilaments and an incomplete B-tubule containing 10 protofilaments (*Warner and Satir, 1973*; *Amos and Klug, 1974*; *Sui and Downing, 2006*; *Nicastro et al., 2011*). Dynein arms are bound to the A-tubule of the dMTs and walk on the neighboring B-tubule. This causes dMTs to slide along each other, introducing sheer, which is

**eLife digest** Some cells have a whip-like appendage called a flagellum. This is most often used to propel the cell, notably in sperm cells, but it can also be involved in sensing cues in the surrounding environment. Flagella are found in all three domains of life—the eukaryotes (which include the animals), bacteria and ancient, single-celled organisms called Archaea—and they perform similar functions in each domain. However, they also differ significantly in their protein composition, overall structure, and mechanism of propulsion.

The core of the flagellum in eukaryotes is made up of 20 hollow filaments called 'microtubules' arranged so that nine pairs of microtubules form a ring around two central microtubules. The core also contains many other proteins, but it is not clear how all these components come together to make a working flagellum. Moreover, it is not known if the flagella of different groups of eukaryotes are all assembled in the same way.

Now, Höög et al. have discovered that although the core structure of the eukaryote flagellum is highly conserved, it can be assembled in markedly different ways. Some species of eukaryote—such as *Chlamydomonas reinhardtii*, a single-celled green alga, and *Trypanosoma brucei*, the protist parasite that causes African sleeping sickness—must grow new flagella when their cells divide, so that each new cell can swim. Using a form of electron microscopy called electron tomography, Höög et al. could see the detailed structure of the growing flagella in three dimensions. At first the cores of the flagella in these two distantly related species grow in the same way. However as the flagella get longer their cores grow in completely different ways. The microtubule filaments in longer flagella grow in a synchronized manner in the alga, but in a disorganized way in the protist.

The results of Höög et al. illustrate that it is not advisable to draw generalised conclusions based on studies of a few model species. However, since defects in flagella are known to cause several diseases in humans, this knowledge might inform future studies aimed at developing treatments for infertility, respiratory problems, and certain kinds of cancer.

converted into flagellar bending by several classes of static links. Flagellar bending is controlled so as to induce motility in some cells (e.g., spermatozoa, *Giardia spp.* and *Trypanosoma spp.*) and propel the surrounding media in other cells (e.g., respiratory tract epithelia, fallopian tube epithelium). In addition to the MT components of the axoneme, partially assembled protein modules such as radial spokes (*Qin, 2004*; *Diener et al., 2011*), nexin links and central pair projections are important for axonemal function through their roles as cross-linkers and regulators of dMT sliding and bending.

In most multicellular organisms, the cilium is produced after the cell has exited the cell cycle, but in many protozoan flagellates, new flagella must be built to maintain motility in daughter cells (*Ginger et al., 2008*; *Dawson and House, 2010*). Flagellar elongation occurs by addition of protein subunits at the axoneme's distal end (*Rosenbaum and Child, 1967*; *Marshall, 2001*). Large protein complexes containing the precursor axoneme building blocks are delivered to this site via an evolutionary con-served process called intraflagellar transport (IFT; [*Kozminski et al., 1993*]). The roles of IFT in ciliary function are well studied (*Pedersen and Rosenbaum, 2008*), and the molecular mechanisms that mediate IFT of axonemal proteins are beginning to be characterized (*Bhogaraju et al., 2013*). The structure of the flagellar tip has been characterized; the B-tubule ends before the A-tubule creating a distal 'singlet region' in the flagellum tip of most species (*Ringo, 1967*; *Satir, 1968*; *Sale and Satir, 1976*; *Woolley and Nickels, 1985*); the CPs extend further into the distal tip than the dMTs (*Ringo, 1967*); the dMTs and CPs are linked to the membrane through capping structures (*Dentler, 1980*; *Woolley et al., 2006*). Yet, we know very little about how the flagellar components, once delivered to the distal tip, are assembled to form the beating flagellum (*Ishikawa and Marshall, 2011*; *Fisch and Dupuis-Williams, 2012*). For example, does the CP extend beyond the dMTs during tip growth, like in the mature flagellum, or is the growth of all MTs synchronized? Alternatively do the dMTs extend beyond the CP during flagellar extension? When do other structural modules such as radial spokes, dynein arms, and central pair projections get incorporated? Clearly, there are multiple possibilities for how a flagellum might extend.

We have examined two evolutionary distant organisms, the green algae *Chlamydomonas reinhardtii* and the parasitic protozoa *Trypanosoma brucei*, to determine if a consistent pattern of flagellar extension

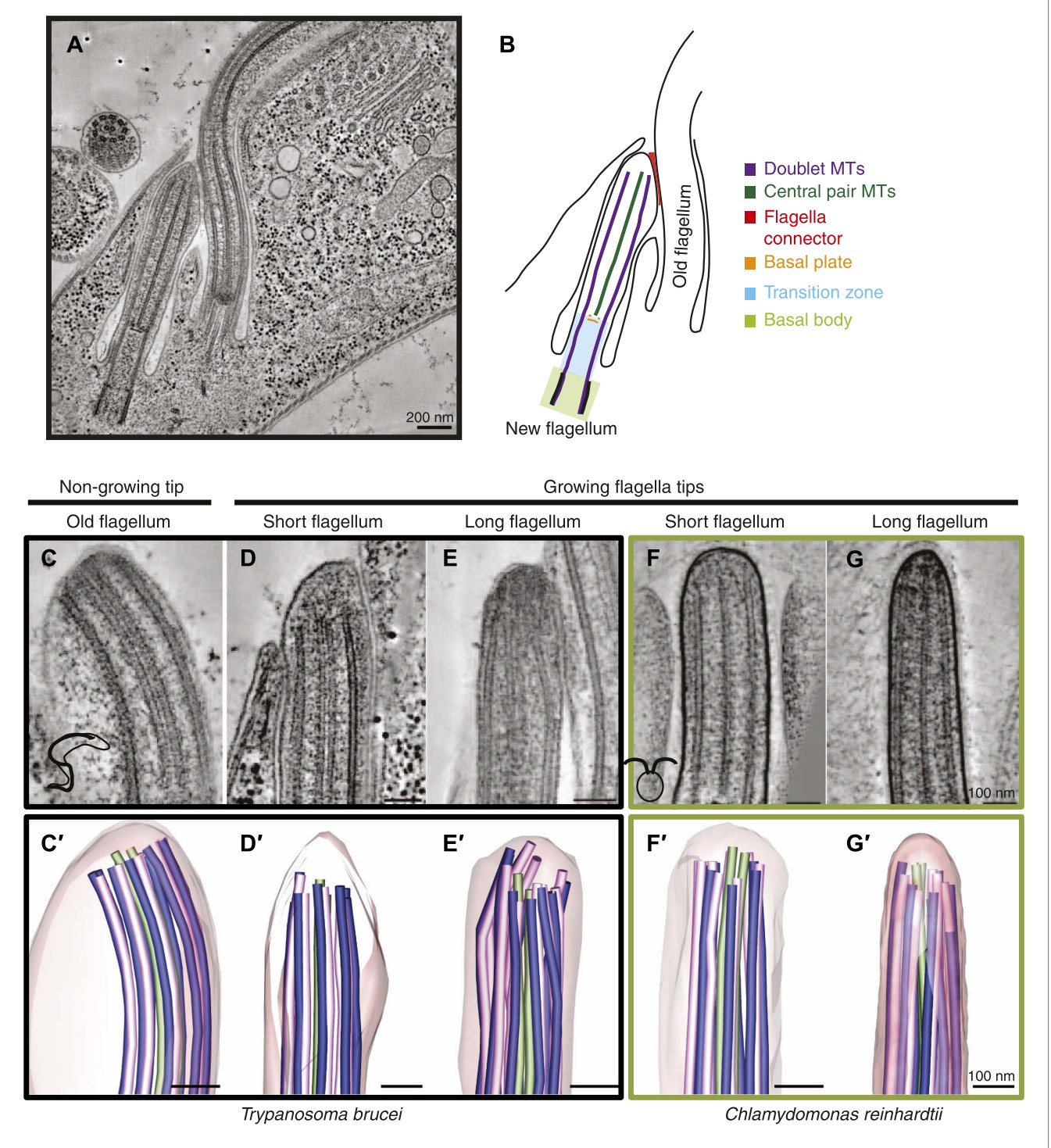

**Figure 1**. Microtubule organization at the distal tip of the flagellum varies in different cell cycle stages and species. (**A**) A 20 nm tomographic slice showing a new *T. brucei* flagellum attached to the old flagellum. (**B**) A schematic of the picture in **A** describing the relevant regions of the flagellum. (**C**–**G**) Tomographic slices (20 nm thick) showing distal tips of flagella in different stages of the cell cycle. (**C**) *T. brucei* old flagellum, (**D**) *T. brucei* short flagellum, (**E**) *T. brucei* long flagellum, (**F**) *C. reinhardtii* short flagellum, and (**G**) *C. reinhardtii* long flagellum. (**C′**–**G′**) 3D models of flagella in the same cell cycle stage as in **C**–**G**, showing the axoneme with A-tubules in pink, B-tubules in dark blue and central pair in green. Flagellar membrane is shown in transparent pink.

The following figure supplements are available for figure 1:

**Figure supplement 1**. Gallery of long growing *T. brucei* tips, all showing disordered axonemes (20 nm thick tomography slices).

**Table 1.** Sample size of each species/cell cycle stage

| | Flagellum type | N (flagella tips reconstructed) |
|---|---|---|
| *T. brucei* | Old | 6 |
| | Short | 5 |
| | Long HPF | 10 |
| | Long chemical fixed | 2 |
| *C. reinhardtii* | Short | 4 |
| | Long | 4 |

exists. By studying the tips of their growing flagella and their basal plate region, we reveal two separate assembly pathways of flagella extension and maturation.

## Results

### Surprising microtubule arrangements at the flagellar tip

To elucidate the pathways for axoneme elongation, we used electron tomography to examine the tips of actively growing flagella in two organisms, just when the flagella have started growing (at ~0.7–1.5 µm), and after a period of flagellar growth (at 4–10 µm; *Table 1*). *C. reinhardtii* has two flagella that are reabsorbed down to their transition zones, which are then expelled prior to mitosis

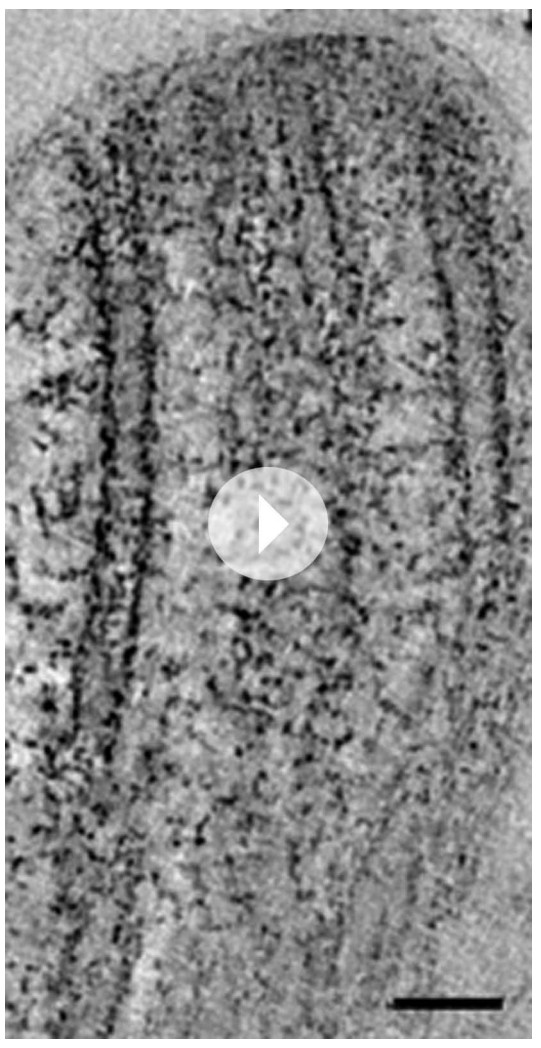

**Video 1**. Old *T. brucei* flagellum tip (related to *Figure 1*). 1-nm thick sections of a tomogram reconstruction containing the distal tip. Scale bar = 50 nm.

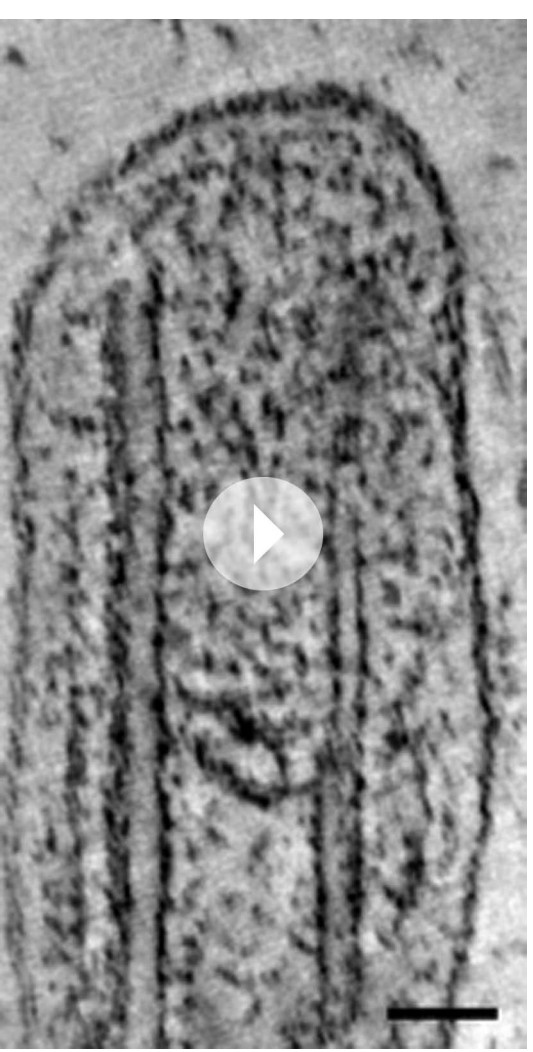

**Video 2**. Growing short *T. brucei* flagellum tip (related to *Figure 1*). 1-nm thick sections of a tomogram reconstruction containing the distal tip. Please also note the just formed basal plate. Scale bar = 50 nm.

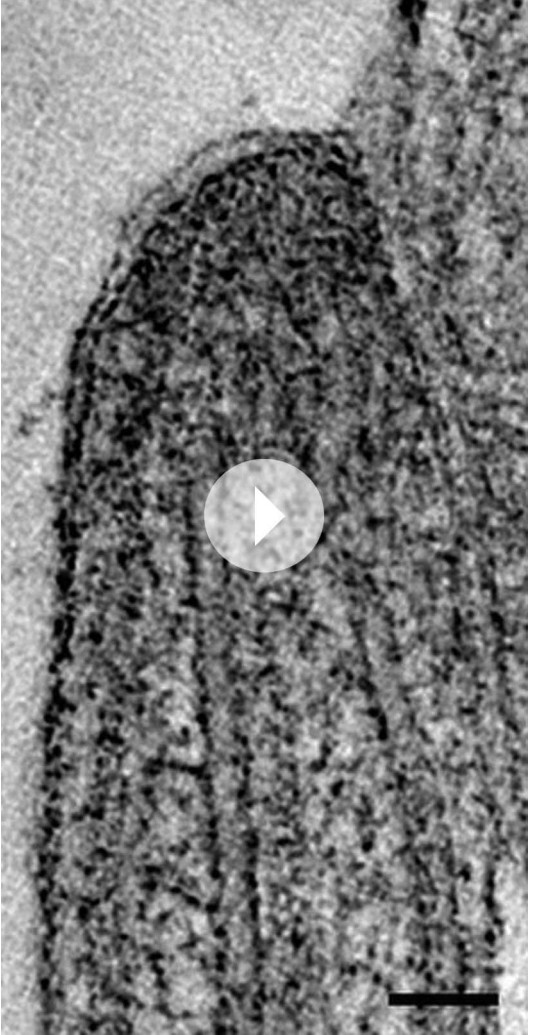

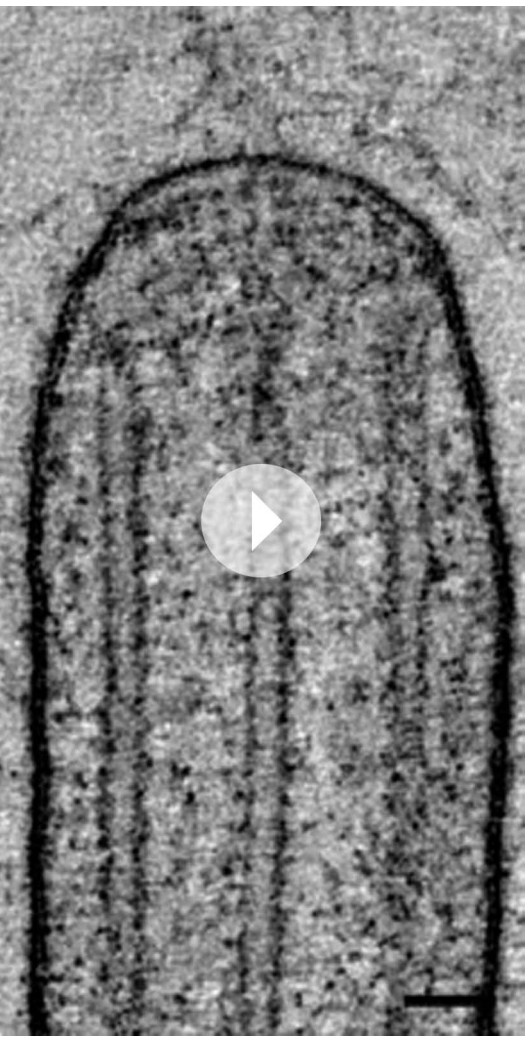

**Video 3**. Disordered long growing *T. brucei* flagellum tip (related to *Figure 1*). 1-nm thick sections of a tomogram reconstruction containing the distal tip. Note the bent doublet microtubules and the absence of electron-dense structures associated with the central pair. Scale bar = 50 nm.

**Video 4**. Short growing *Chlamydomonas reinhardtii* flagellum tip (related to *Figure 1*). Scale bar = 50 nm.

(*Rasi et al., 2009*; *Parker et al., 2010*). After mitosis, the small daughter cells remain within the wall of the mother cell where they regrow their flagella; a cell stage we easily identified in the electron microscope. In *T. brucei* the new flagellum starts growing midway through the cell cycle (*Sherwin and Gull, 1989*). Its tip is attached to the side of the old flagellum by a structure called the flagella connector (*Moreira-Leite, 2001*; *Briggs, 2004*; *Davidge, 2006*). The relative positions of a cell's two flagella allow an unambiguous identification of which flagellum is old and which is new (*Figure 1A,B*). Throughout the rest of the cell cycle, the length of the new flagellum correlates with cell cycle progression.

3D electron tomographic reconstructions of flagellar tips were made of old *T. brucei* flagella (slowly growing or of constant length; *Video 1*) (*Davidge, 2006*; *Farr and Gull, 2009*), and growing new flagella that were short or long (*Videos 2 and 3*). Comparable images of new growing flagella that were short or long were also obtained from *C. reinhardtii* (*Videos 4 and 5*). The axoneme in all flagellar tips studied, except the *T. brucei* growing long flagella, displayed the regular spacing of dMTs found in the rest of the flagellum (*Figure 1C,D,F,G*, *Figure 1—figure supplement 1*). In contrast, the growing long *T. brucei* flagellar tips showed a disorganized array of dMTs (*Figure 1E*); some dMTs lay very close to the CPs. The axoneme structure is revealed in 3D models of each of the tips (*Figure 1C′–G′*, *Videos 6–10*). The disorganized microtubules at the tips of growing flagella in *T. brucei* indicate that the mechanism of

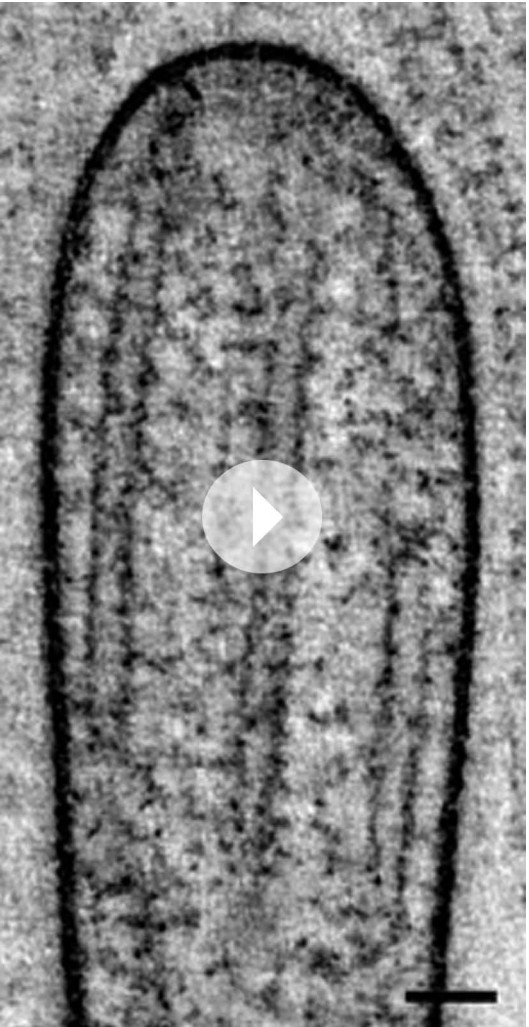

**Video 5**. Long growing *Chlamydomonas reinhardtii* flagellum tip (related to *Figure 1*). Scale bar = 50 nm.

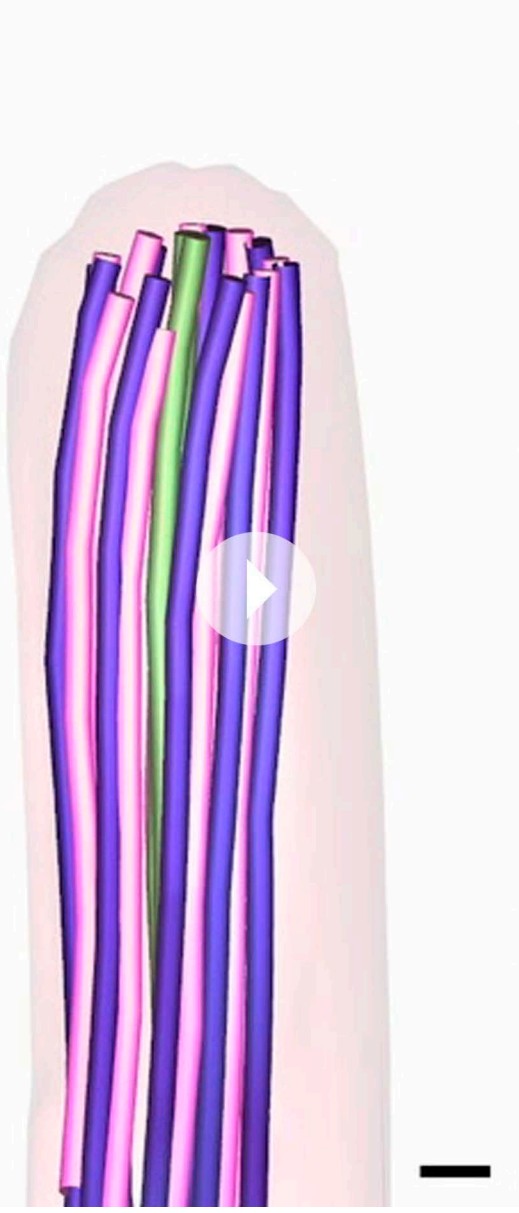

**Video 6**. 3D model of the microtubules found in an old *T. brucei* flagellum tip (related to *Figure 1*). Model was made by drawing lines in the microtubules and around membranes as seen in a tomogram reconstruction of the old *T. brucei* flagellum tip (*Video 1*). The lines were then provided with a skin through a meshing process. Scale bar = 50 nm.

axonemal growth in long flagella is different in this species from that seen in *C. reinhardtii*, so we investigated the structure of growing flagella further.

## Missing axoneme-associated structures at the growing tip of *T. brucei* flagella

The structural disorder in growing long new flagella of *T. brucei* is most obvious in 3D models generated by tracing the doublet MTs through tomographic reconstructions (*Figure 2A,B*). ~300 nm from the tip of a *T. brucei* flagellum, the nine doublet MTs make an almost perfect circle around the central pair MTs (*Figure 2B3*). However, ~150 nm from the tip, the MT doublets change orientation compared to the central MT pair (*Figure 2B2*). From this point toward the axoneme tip, the MT doublets lose their circular arrangement (*Figure 2A–B1*). In this disordered region, some MTs lie closer to the CPs, others are further away from them (*Figure 2C*, *Figure 1—figure supplement 1*), indicating that the tip is not just a tapered end.

The disorder seen in the growing *T. brucei* axoneme could be explained by a transient lack of MT-associated structures that are present in the mature axoneme, for example the radial spoke

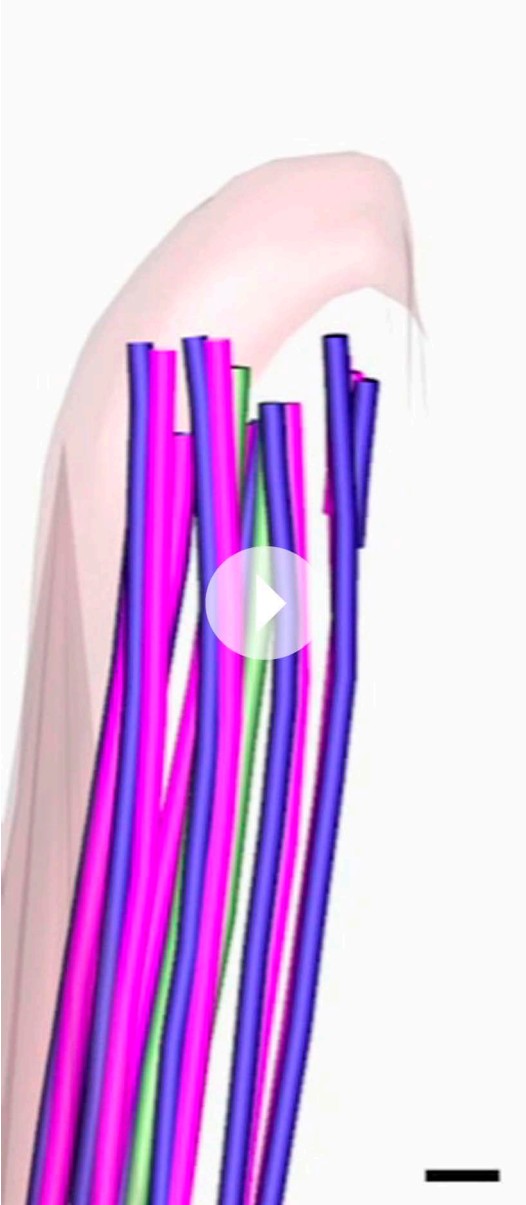

**Video 7**. 3D model of the microtubules found in a short growing *T. brucei* flagellum tip (related to *Figure 1*). Model is a segmentation from the tomogram, *Video 2*. The tomogram reconstruction contained the most of the flagellum tip, but cropped off some the axoneme shortly proximal to it and one dMT was not found within the reconstruction (missing in the model). Scale bar = 50 nm.

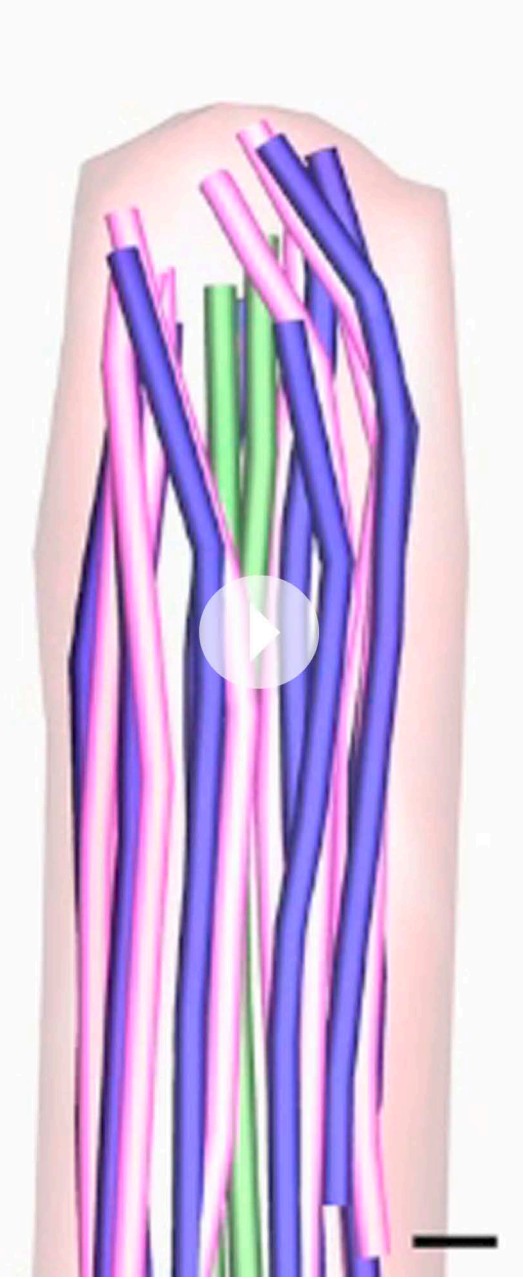

**Video 8**. 3D model of the disordered microtubules found in a growing *T. brucei* flagellum tip (related to *Figure 1*). Note that some doublet microtubules are in contact with the flagella membrane. Scale bar = 50 nm.

complex, CP projections and/or nexin links. Central pair projections display a ladder-like arrangement along the length of the mature *T. brucei* flagellum (*Figure 2D*), and in all the distal flagella tips (*Figure 2E,F,H–J*) except for in the growing *T. brucei* long flagellum tip (*Figure 2G*), where the associated complexes were not visible. Therefore, although we have a relatively small sample size (*Table 2*), we suggest that during the elongation of a *T. brucei* flagellum, the microtubules extend beyond the position of the associated proteins, and hence the final structures are added to the already formed but disorganized axonemal tip.

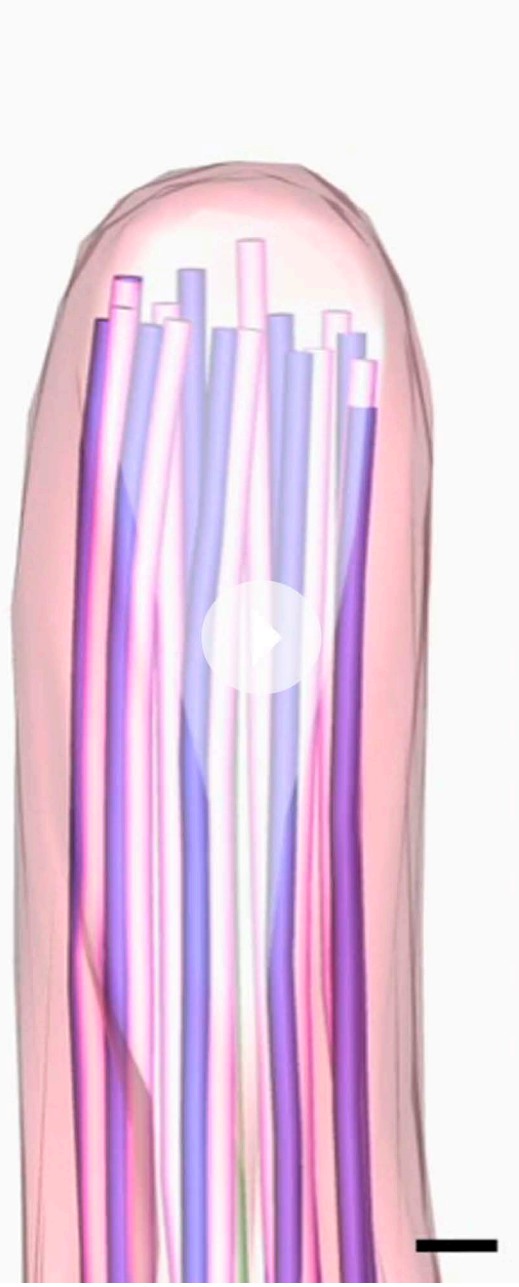

**Video 9**. 3D model of a short growing *C. reinhardtii* flagellum (related to *Figure 1*). Scale bar = 50 nm. DOI: 10.7554/eLife.01479.014

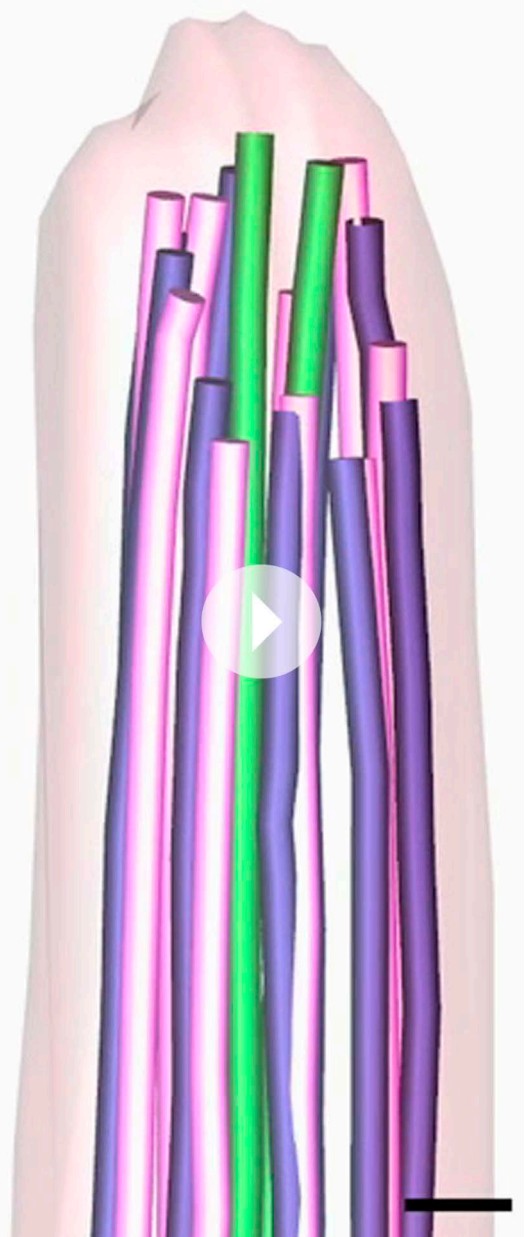

**Video 10**. 3D model of a long growing *C. reinhardtii* flagellum (related to *Figure 1*). Scale bar = 50 nm. DOI: 10.7554/eLife.01479.015

## Axonemal microtubule and flagellar membrane extension

We investigated whether the plus ends of axonemal MTs pushed against the flagellar membrane as they grew. The 3D nature of our data allowed the quantification of microtubule end distances to the nearest flagellar tip membrane. The reconstructions of old *T. brucei* flagella showed dMTs as well as the CP plus ends neatly arranged ~30 nm from the tip membrane (*Figure 3A*). Only in the disorganized growing long *T. brucei* flagella did dMTs sometimes touch the flagellar membrane (distance 0–125 nm; average 44 ± 32 nm; n=36 dMTs; *Table 2*). In both short and long growing *C. reinhardtii* flagellum tips, all axonemal MT ends were found within 230 nm of the nearest flagellar tip membrane (*Table 2*). In no case did the CPs touch the flagellar tip membrane. Thus, in all cases except growing long *T. brucei* flagella, flagellar extension does not require close contact between dMTs and flagellar membrane.

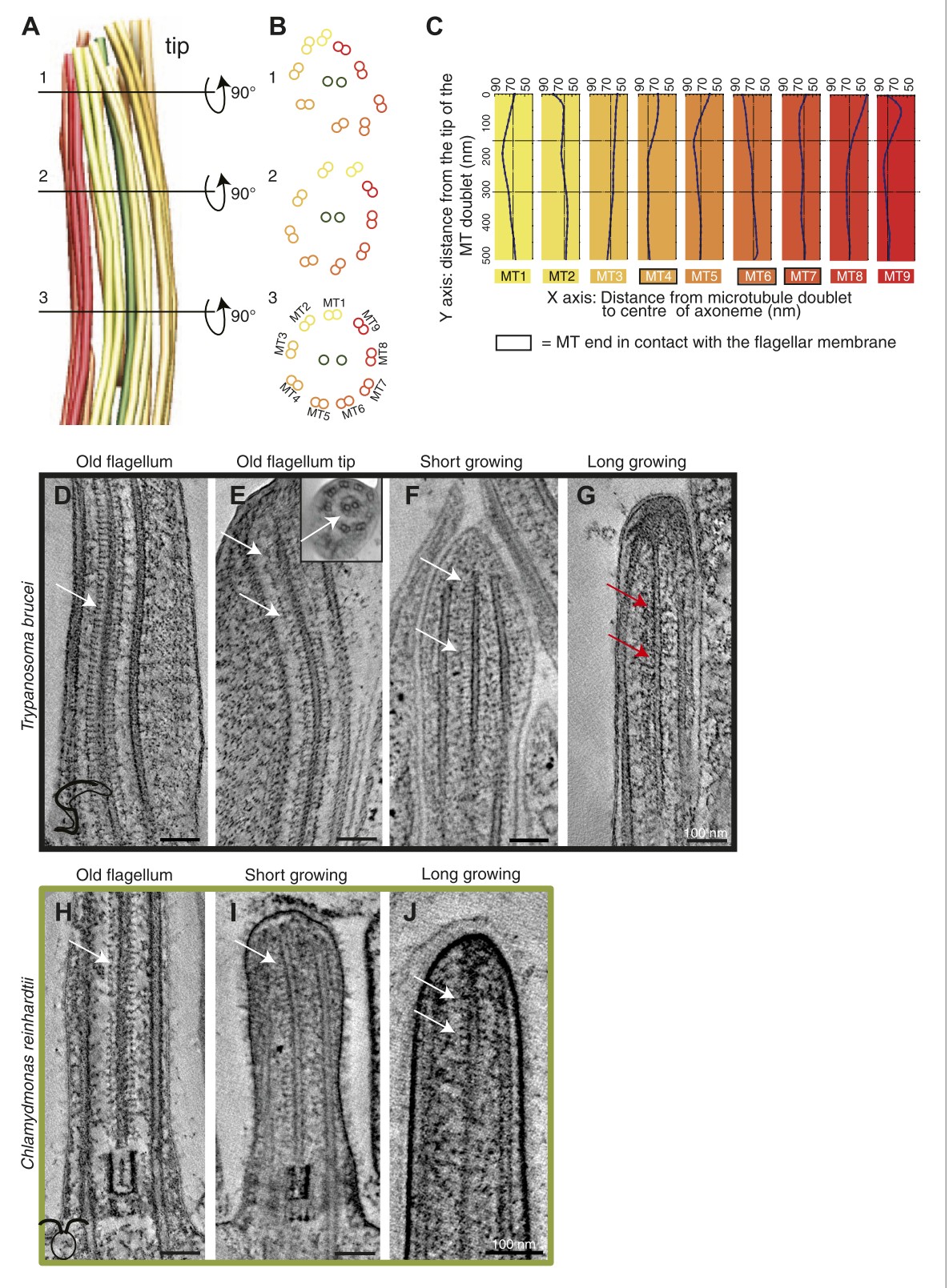

**Figure 2**. Structural disorganization in the growing tips of long *T. brucei* flagella. (**A**) 3D model of a tomographic reconstruction of a growing, long *T. brucei* axoneme. The doublets are color coded in a gradient from yellow (dMT1) to red (dMT9). (**B**) A cut through of the 3D model shows that (1) at the end of the flagellum, the circular arrangement of the MTs is completely lost, (2) <0.5 μm before the flagella tip, the MTs start losing their circular

*Figure 2. Continued on next page*

*Figure 2. Continued*

organization, and (3) >0.5 µm from its tip the axoneme is well organized. (**C**) Individual traces of the nine dMTs in **A** reveals them bending both toward and away from the CP. dMTs not touching the membrane also showed this random bending (e.g., dMTs 2 and 5) (**D**) CP projections are arranged as an electron-dense ladder (arrows) extending from the central pair. (**E**) In the tip of the *T. brucei* old flagellum the associated structural proteins are visible all the way to the tip. In the longitudinal view, we see the distal end that then curves and the final 200 nm is shown in cross-sectional view (insert). (**F**) Central pair projections are clearly seen all the way to the end of the central pair in the short growing *T. brucei* flagellum. (**G**) In the *T. brucei* long growing flagellum such associated proteins are not visible (red arrows). (**H**) The structural proteins are clearly visible along the length of a mature *C. reinhardtii* flagellum. (**I** and **J**) In the growing short *C. reinhardtii* flagellum, these proteins were also present at the axoneme's tip, indicating that microtubules and associated structures are simultaneously assembled.

We also investigated whether there was a difference between the elongation of CPs and dMTs. To show microtubule extension within the axoneme, we measured the distance from the ends of the A-, B-tubule and CP to the end of the furthest reaching microtubule of the axoneme. The ends of dMTs and CPs of flagellar ends did not significantly differ in their extension in most samples (*Figure 3B*; *Table 3*). However, we found a significant difference between the extensions of dMTs and CPs in the growing long *T. brucei* flagella, where the CP lagged behind the dMTs extension by approximately 50 nm (p<0.01). Also in *C. reinhardtii*, a significant difference was revealed, but here the CP extended ~50 nm beyond most dMTs (p<0.01).

The small distance from the tip in which all the MT tips were found in both *T. brucei* and *C. reinhardtii* contradicts the existence of a long singlet region, previously published to be ~1 µm in *C. reinhardtii* (*Ringo, 1967*; *Satir, 1968*; *Sale and Satir, 1976*; *Woolley and Nickels, 1985*). Furthermore, the extension of the A- and B-tubule within the dMTs revealed that the A-tubule commonly extended the furthest (*Figure 3C*). However, the extension of the longest sub-fiber did not protrude more than 10–20 nm beyond its partner tubule. No samples showed a significant difference between A- and B-tubule extension (*Figure 3D*; *Table 3*).

We also compared the distances from the MT ends to the tip membrane in high pressure frozen vs chemically fixed growing *T. brucei* flagella (*Figure 3—figure supplement 1*). In chemically fixed cells, the dMT and CP ends were found in a wide range of distances, up to 350 nm from the membrane (average 203 ± 75 nm; n = 40), in contrast to the high pressure frozen *T. brucei* flagella where the same distance was a maximum of 125 nm (45 ± 37 nm; n = 75). The microtubule plus ends within the same axoneme were spread over a larger distance in chemically fixed samples than in high pressure frozen ones. Indeed, a partial axoneme was previously used to localize an image as having been acquired close to the distal flagellum end (*Briggs, 2004*), when such an area of a partial axoneme was rarely to be found in this study. These findings, in combination with the absence of a singlet region in high-pressure frozen samples, indicate that the distal flagellar arrangements previously published in *C. reinhardtii* tips were likely disturbed by the chemical fixation used.

We conclude that the singlet region is not found in either species examined, but that the two species examined display two different assembly pathways.

## Microtubule plus end anchoring

Axonemal microtubules have previously been shown to be linked to the flagellar membrane (*Dentler, 1980*; *Woolley et al., 2006*). We examined the presence of such linkages in all 3D reconstructions

**Table 2.** Distances between the axonemal microtubules and the flagellar membrane

| | Flagellum type | N (A/B-tubule and CPs) | Average MT-membrane distance (nm) | Range: MT-membrane distance (nm) |
|---|---|---|---|---|
| *T. brucei* | Old | 48 | 26 ± 9 | 9–54 |
| | Short | 54 | 94 ± 27 | 35–160 |
| | Long | 47 | 50 ± 37 | 0–157 |
| *C. reinhardtii* | Short | 60 | 77 ± 32 | 33–214 |
| | Long | 60 | 97 ± 47 | 42–230 |

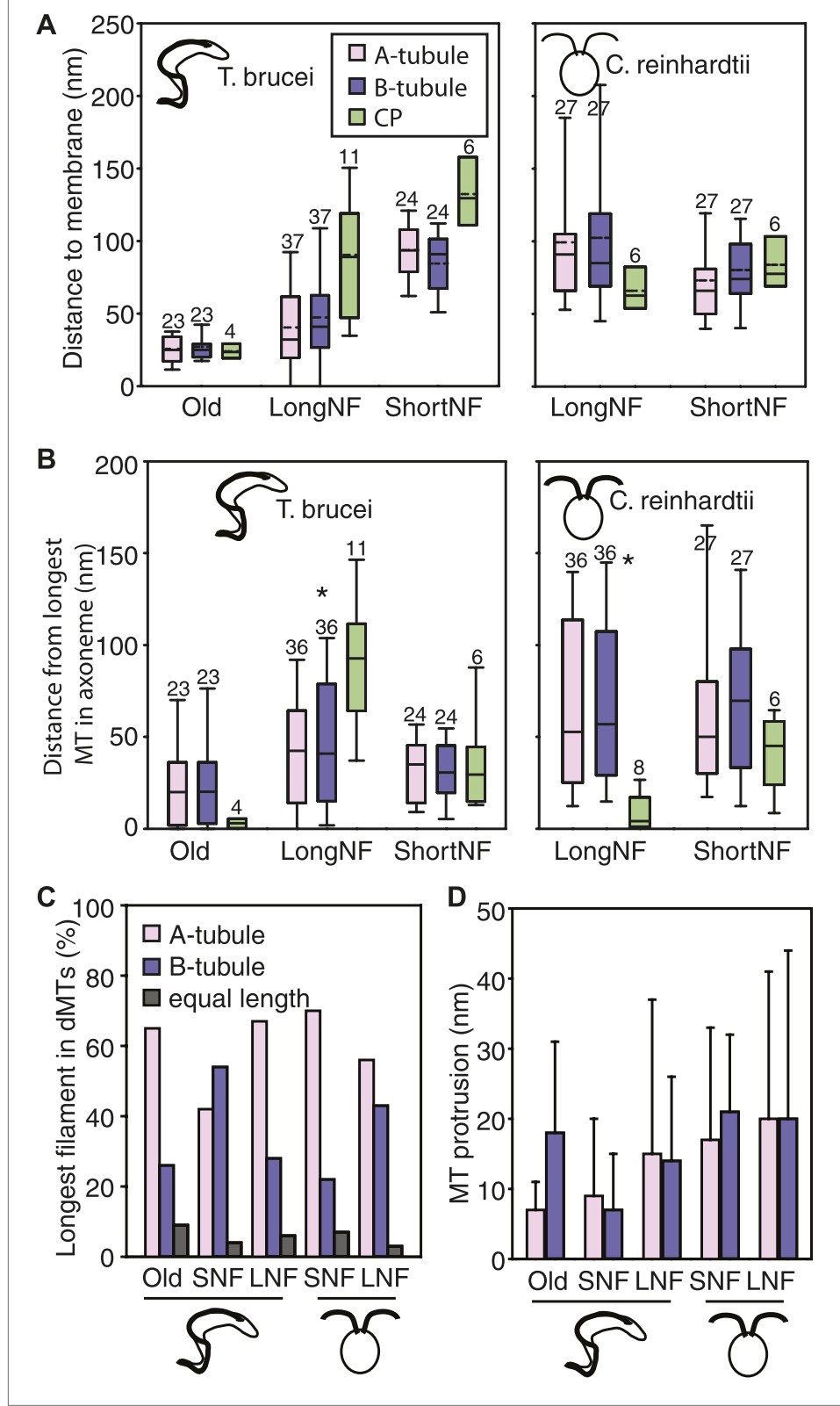

**Figure 3.** No singlet zone in any of the samples examined. (**A**) The distance between the dMTs/CP MT plus ends and the closest flagellar tip membrane was measured in flagella of each cell cycle stage/species. These properties are represented here as box plots in which the mean value is identified by a dotted horizontal line,
*Figure 3. Continued on next page*

*Figure 3. Continued*

the median value by a line, the 25–75% percentile by the height of the box, and the horizontal bars outside the box include the 5th and the 95th percentile. Only in *T. brucei* long new flagellum (LNF) were the microtubules touching the membrane, but in all samples all MT ends were closer than 230 nm to the membrane. (number above box = n) (**B**) To determine if dMTs or CP microtubules extended the furthest within the axoneme, all microtubule plus end's distance to the longest microtubule in the axoneme was measured. (*) In *T. brucei* LNF, the CPs was lagging behind the dMT extension, and in *C. reinhardtii* CPs extended further than the dMTs. (**C**) The extension of the A- or B-tubule within the doublet was measured. The A-tubule did not always extend the furthest. (**D**) The difference of extension between the A- and B-tubule in a dMT is merely 10–20 nm in both species.

The following figure supplements are available for figure 3:

**Figure supplement 1**. Chemical fixation increased the distance between the flagellar tip membrane and the microtubule plus ends.

---

available (*Figure 4A–H*). The distal ends of the flagella were often very electron dense, particularly in the growing long flagella (*Figure 4C,G*), making visualization of the MTs and their ends difficult. Most MT with a clear end morphology appeared flared, but in some cases caps were visible on CPs (e.g., *C. reinhardtii* short new flagellum; *Figure 4E*, middle). We saw some filamentous material between the CPs and dMTs extending to the flagellar membrane, and also from the CP to the dMTs (e.g., *Figure 2J*) but it is not a clear structural link. We saw no evidence of a bead and plate structure, as previously described in the *C. reinhardtii* CP cap found in demembranated mature flagella (*Dentler and Rosenbaum, 1977*). In cryo-electron tomography of an intact *T. brucei* old flagellum tip, we see clear electron densities in the distal end of dMT A-tubules as well as in CPs, but no bead or plate structure (*Figure 3I,J*). Thus, the distal ends of axonemal microtubules are likely linked to the membrane by fibrous structures in the mature flagellum, but our results are inconclusive about the growing axoneme.

## Maturation of the flagellum's proximal end in *T. brucei*

We investigated whether the proximal end of the flagellum becomes altered as the flagellum grows and matures. Electron tomographic reconstructions of five cells revealed the ultrastructure of the basal plate and the anchoring of the central pair microtubules throughout the cell cycle in ten flagella (*Figure 5A*; five new and old flagellum pairs).

Several differences were found over the cell cycle in *T. brucei*: first, the minus ends of the central pair microtubules were capped in all four growing new flagella long enough to have a CP (*Figure 5B* and arrow in D; *Video 11*), whereas in old flagella of the same cells, CPs showed some open ends (6 out of 9 CP ends of known morphology; *Figure 5B* and arrow in G; *Video 12*). Second, in new flagella, the ~30-nm thick basal plate consisted of two stacked electron-dense rings, and the minus ends were found within the more distal ring (*Figure 5C,D*, *Figure 5—figure supplement 1*). In old flagella, this defined basal plate structure consisting of two rings was gradually lost and replaced with a more diffuse electron dense mass that became up to 90 nm thick with the progression of the

**Table 3.** Statistics on the microtubule extension within the axoneme

| | Flagellum type | dMT/Cp extension difference | Paired *t* test | n (axonemes) | A-B tubule extension Difference | Paired *t* test | N (MTs) |
|---|---|---|---|---|---|---|---|
| *T. brucei* | Old | No | 0.24 | 2 | No | 0.63 | 23 |
| | Short | No | 0.81 | 3 | No | 0.73 | 24 |
| | Long | Yes | 0.01 | 6 | No | 0.12 | 36 |
| *C. reinhardtii* | Short | No | 0.28 | 3 | No | 0.14 | 28 |
| | Long | Yes | 0.01 | 4 | No | 0.51 | 36 |

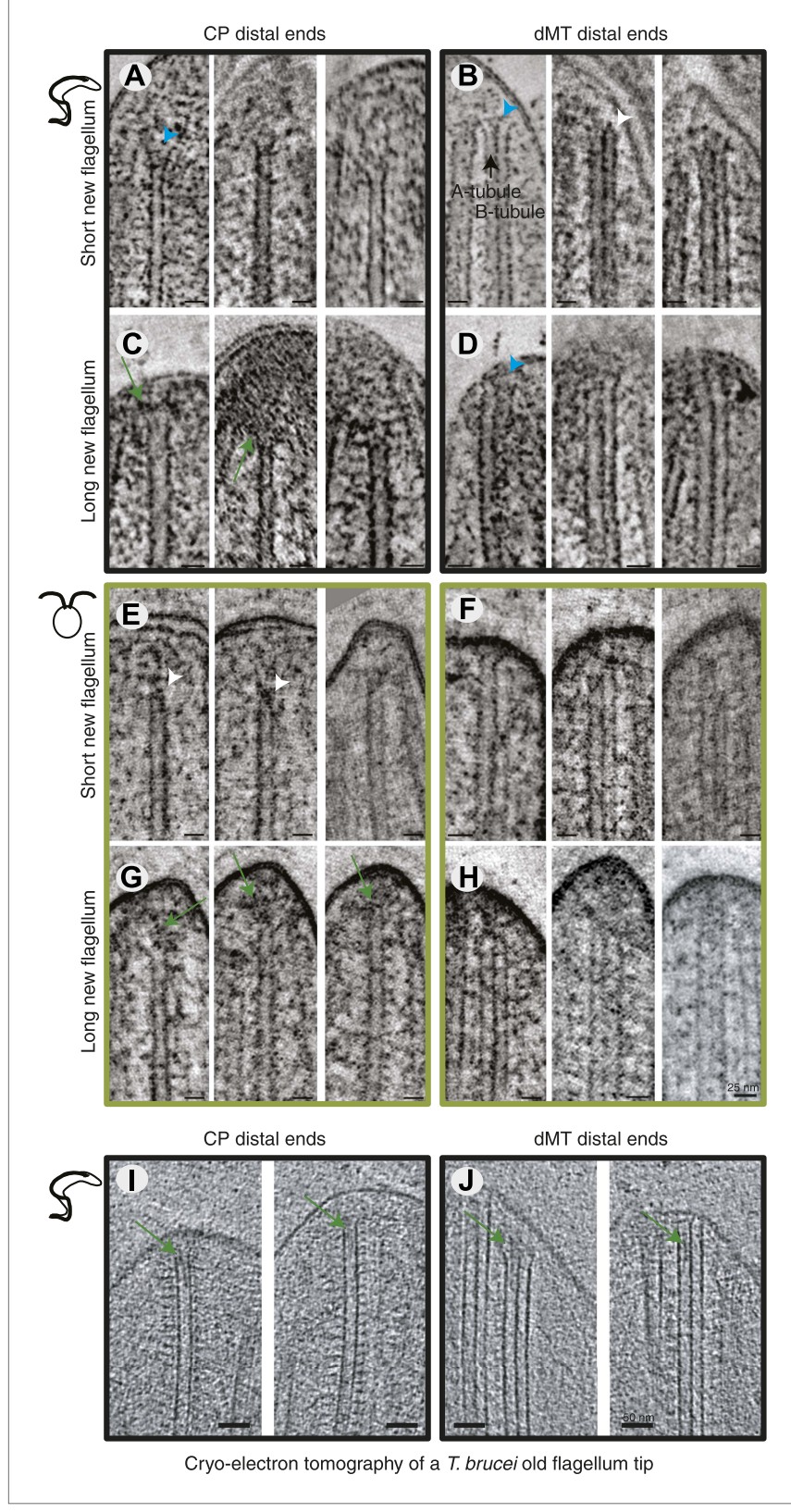

**Figure 4**. All axoneme microtubule plus ends are found close to the tip membrane. 4-nm thick slices of tomograms from (**A** and **B**) *T. brucei* short new flagellum, (**C** and **D**) *T. brucei* long new flagellum, (**E** and **F**) *C. reinhardtii* short new flagellum, and (**G** and **H**) *C. reinhardtii* long new flagellum. In the left column CP distal ends are shown and the
*Figure 4. Continued on next page*

*Figure 4. Continued*

right column, dMT plus ends. dMTs are always displayed with the A-tubule to the left. Examples of flared ends are marked with turquoise arrowhead and capped ends with white arrowheads. Electron-dense structures associated with the *C. reinhardtii* and *T. brucei* CPs in long new flagella are marked by green arrows. (**I** and **J**) 15-nm thick cryo-electron tomography sections of (**I**) CP and (**J**) dMTs at the old *T. brucei* flagellum tip. Note the electron density extending into the lumen of both CP MTs and into the A-tubule lumen in the dMTs.

flagellum maturation (*Figure 5B,G,H*). Third, the central pair minus ends were commonly found within this electron dense mass in old flagella, but in 2 out of the 10 CPs, one of the ends was found around 100 nm distal to the basal plate (*Figure 5B,G*; cells 3 and 5), showing a loosened CP anchoring in more mature flagella.

Thus, the *T. brucei* basal plate structure and its association with the central pair minus ends changes as the flagellum matures. We wondered if this is a conserved feature of the CP nucleating region of flagella (*Euteneuer and McIntosh, 1981*; *Song and Mandelkow, 1995*).

## Little structural maturation in the *C. reinhardtii* transition zone

We therefore reconstructed nine transition zone regions of *C. reinhardtii* cells found within different mother cell walls (*Figure 6A*). These cells had flagella of various lengths, all growing except for two flagella that were found in a mature cell (shown as 'long'). We first plotted the thickness of the central cylinder of the transition zone (previously described as an electron dense H and the core of the 9-pointed star [*Ringo, 1967*]) against the length of the flagellum to see if there were any obvious changes to this structure as the flagellum grows. The nine central cylinder structures were all between 120 and 200 nm (average 158 ± 25 nm), but there was no detectable increase in their thickness with flagella length. The assumed CP minus ends were in close proximity to the distal end of the cylinder, and 14 out of 17 minus ends were capped. The open CP ends were found in rather short flagella (1.2 and 2 µm; *Figure 6B*; *Video 13*).

Three cells were arranged by the lengths of their flagella and thus, their cell cycle stages (*Figure 6C–E*). The central cylinder looks like two stacked U's in tomographic slices of all cells (*Figure 6D*). Density thresholding of our tomograms reveals that the U's resemble two stacked baskets in 3D (*Figure 6E*). As the central cylinder grows, the proportions of the baskets remain surprisingly similar (*Figure 6D,E*), with the longer upper basket constituting 60 ± 5% of the total basal plate thickness and the more partial lower basket constituting 28 ± 5% (*Figure 6F*). Interestingly, in the one cell with long flagella where we reconstructed both transition zones, the two central cylinder structures were similar in thickness and in their anchoring of the CPs (*Figure 6A,B*; long flagellum).

We conclude that the structural alterations in the region of CP nucleation during flagellar maturation in *T. brucei* are not conserved between these two species, as the *C. reinhardtii* structure merely grows lengthwise.

## Growth and maturation of the flagellum

Since the flagellum extends at the distal end, one has a timeline of flagellar extension with the most recently built piece at the distal tip and the oldest region at the proximal end (*Figure 7A,B*). We used this property to dissect the assembly process in the disorganized growing long *T. brucei* flagellum and found the central pair projections (shown to be missing in these tips in *Figure 2F*) ~0.5 µm from the distal tip (*Figure 7C*). However, already after 0.25 µm the axoneme had its normal circular arrangement, which was also where the radial spokes were incorporated into the axoneme, confirming the crucial role of the radial spokes in the circular arrangement of dMTs of these flagella. Interestingly, it was also at ~250 nm from the distal tip that the microtubules were found to start in chemically fixed cells (*Figure 1—figure supplement 1*), suggesting that the presence of radial spokes stabilized the dMTs and prevented further shrinkage. The paraflagellar rod (PFR), an extra-axonemal para-crystalline structure found in *T. brucei* and many other kinetoplastids (*Vickerman, 1962*; *Bastin et al., 1998*; *Portman and Gull, 2010*; *Höög et al., 2012*), was the last component added to the flagellum, ~800 nm from the tip.

Finally, we used this information to build a summarizing model of the two ways to grow and mature a flagellum in the protozoa *T. brucei* and *C. reinhardtii*. In *T. brucei* the growing axoneme is disorganized, with shorter CPs than dMTs, all of which are possibly not anchored to the flagellar membrane (*Figure 7D*). The associated axonemal protein modules such as radial spokes and CP projections

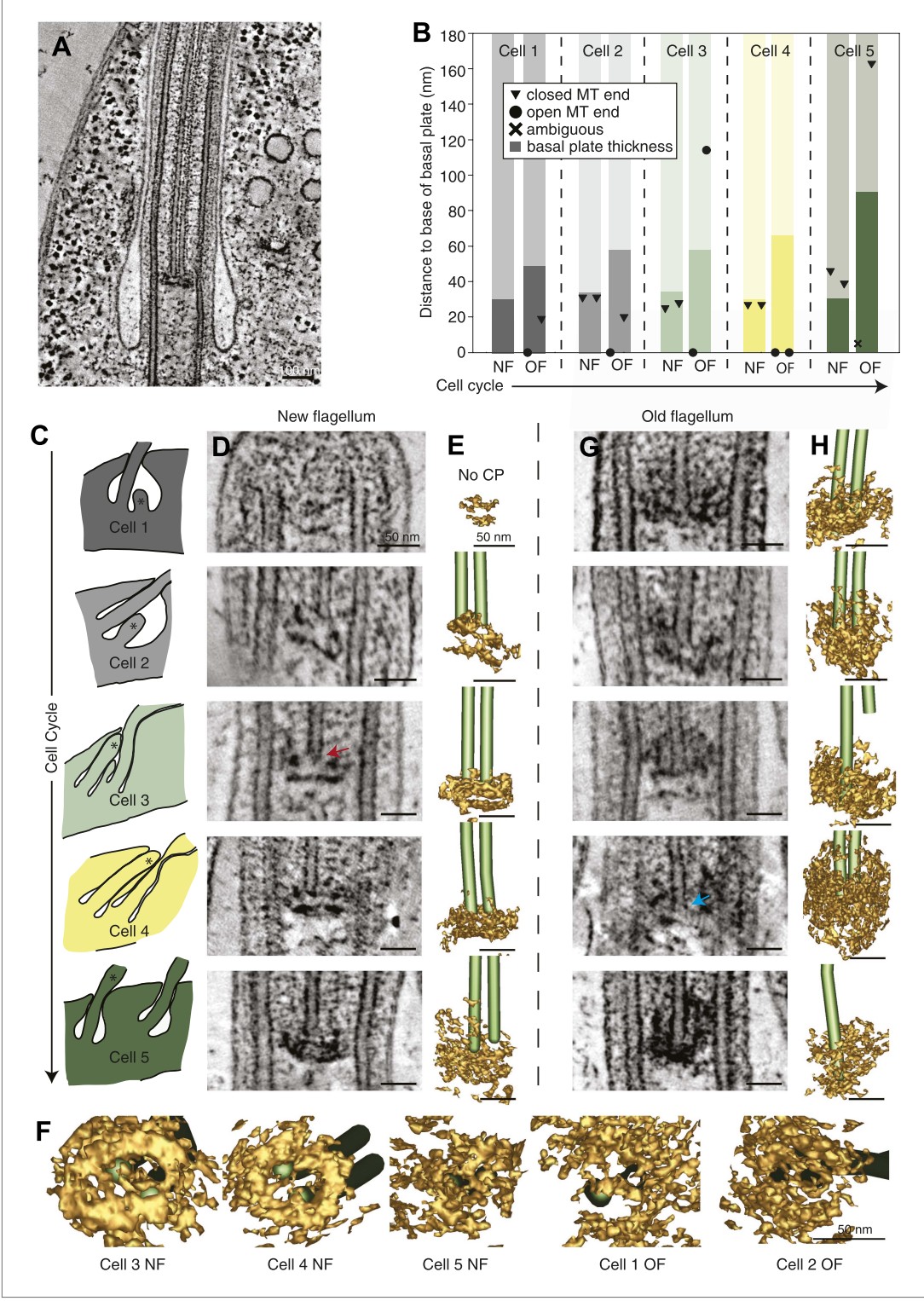

Figure 5. The *T. brucei* basal plate matures and alters its association with the CP minus ends during the cell cycle. (**A**) A 20 nm thick tomographic slice showing the new flagellum (NF) basal plate region. (**B**) The exact thickness of the basal plates and the locations and structure of the CP minus ends within/around them. The new flagellum in Cell1 is so short that no CP microtubules have started to grow yet. (**C**) The length of the new flagellum (marked with *) reveals the cell's position in the cell cycle; The cell earliest in the cell cycle displaying a short new flagellum (top of Figure) and the most mature cell with two flagella in individual flagellar pockets at the bottom. The cartoons to the

*Figure 5. Continued on next page*

*Figure 5. Continued*

left show outlines of the flagellar pocket region in the cells studied, the cartoons are oriented so that the anterior end of the cell points to the right. (**D**) 10-nm thick tomographic slices of the new flagella showing the bilayered electron-dense material forming the basal plates, with the capped minus end (e.g., turquoise arrow) of one of the CP MTs nucleated on the proximal surface. Cell 1 has such a short flagellum that no CPs has grown yet, but an early basal plate is evident. (**E**) The CP MTs were modeled and the basal plates visualized using electron density thresholding. The bi-layered structure of the basal plate is clearly visible until cell 4, where the two flagella have separated into two separate flagellar pockets in preparation for cell division. (**F**) A selection of the models displayed in **D** and **F** show how the basal plate is formed of two stacked rings early in the cell cycle, which is then lost as the flagellum matures. (**G**) The tomographic slices of basal plates in the old flagella (OF) of the same cell as the new flagellum shown to the left. The bilayered structure is mostly lost, the basal plate is longer and the CP microtubule now has an open end (e.g., red arrow) i.e. inserted further into the basal plate. (**H**) The 3D models of the basal plate region in the old flagella reveal amorphous structures that sometimes only anchor one CP MT (cell 3 and 5).

The following figure supplements are available for figure 5:

**Figure supplement 1**. The ring structure of the new flagellum basal plate in *T. brucei*.

are added in later assembly steps. In the proximal part of the growing flagellum, the basal plate is first arranged into two electron-dense rings close to which capped CP ends can be found. In the corresponding area of a mature flagellum, the basal plate is a loose electron-dense area in which one, sometimes two, open or closed CP minus ends are found.

In *C. reinhardtii*, the growing axoneme is not very different (on this ultrastructural level) from the mature axoneme as the circular arrangement, with the associated axonemal proteins, is maintained all the way to the growing tip (***Figure 7E***). This holds true also for the central structure of the transition zone, the only maturation of which seems to be a lengthwise extension.

## Discussion

Despite the remarkable conservation of the axonemal structure, we have shown two ways to build a flagellum; flagellar growth occurs in both species and length-dependent manner. Whereas, we see no difference in growth in short *T. brucei* and *C. reinhardtii* flagella, *T. brucei* long flagella appear to first elongate their axonemal MTs in a disordered manner, which is then stabilized to a circular arrangement by the addition of associated protein complexes such as the radial spokes. In contrast, our study suggests that the axonemal dMTs and structural proteins are assembling simultaneously in long *C. reinhardtii*. What regulates such differences in the assembly pathways? ***Dentler (1980)*** showed that the growing axoneme in *C. reinhardtii* had structures linking from the tips of the microtubules to the membrane. ***Woolley et al. (2006)*** failed to see such linking structures in growing flagella of *Leishmania major* (but did see them in mature flagella), a close relative to *T. brucei*. We speculate that the presence of such anchorage in *C. reinhardtii* could provide structural clues for the growing axoneme, holding it into a close-to-circular arrangement. This arrangement could then facilitate the incorporation of the associated axonemal proteins such as radial spokes.

Another explanation for the apparent difference in axonemal growth mechanism between *T. brucei* and *C. reinhardtii* could be that these flagella are elongating at different rates: snapshots of a fast-growing axoneme might capture disordered intermediates that are not seen in a more slowly elongating structure. However, the initial growth rate of regrowing *C. reinhardtii* flagella (shed by ionic shock) is approximately 12–24 μm/hr, which then declines to 9 μm/hr as the flagellum gets longer (***Rosenbaum et al., 1969***; ***Flavin and Slaughter, 1974***). The growth rate of their flagella after mitosis remains unknown, but we assume equal or faster assembly because of the pre-mitotic reabsorbtion of the flagellar components that could then be utilized in the construction of the next flagellum. In *T. brucei*, we deduced the flagellum growth rate from the growth rate of the PFR to be a constant ~4 μm/hr (***Bastin et al., 1999***). These growth rates indicate that the structural differences we see are not caused by a slower flagellar growth rate, and therefore more organized growth, in *C. reinhardtii*.

The differences in growth mechanics may not be so surprising, since the two species grow their flagella in different circumstances. In *T. brucei*, the new flagellum is not necessary for cell motility since the old flagellum is still present and active. In *C. reinhardtii* on the other hand, both flagella have been

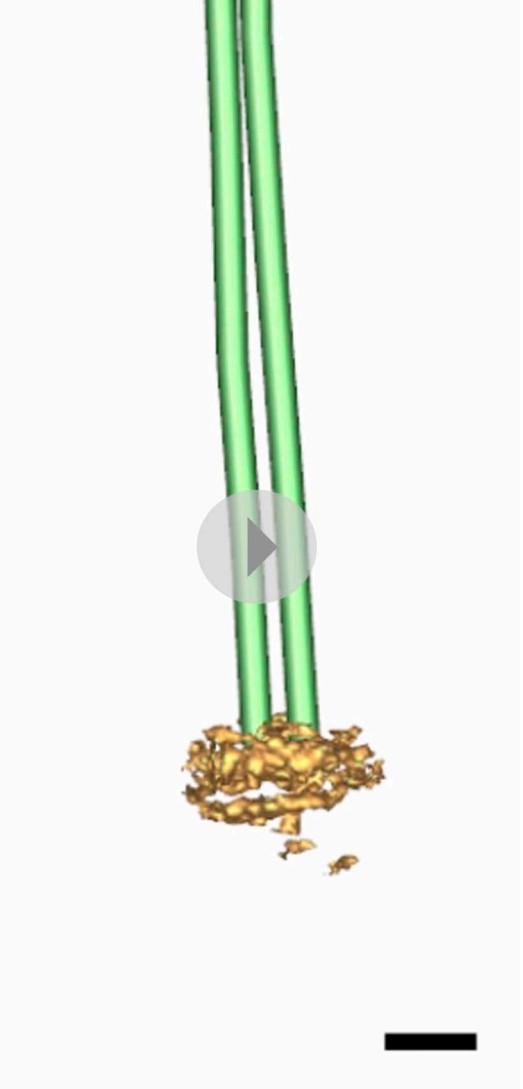

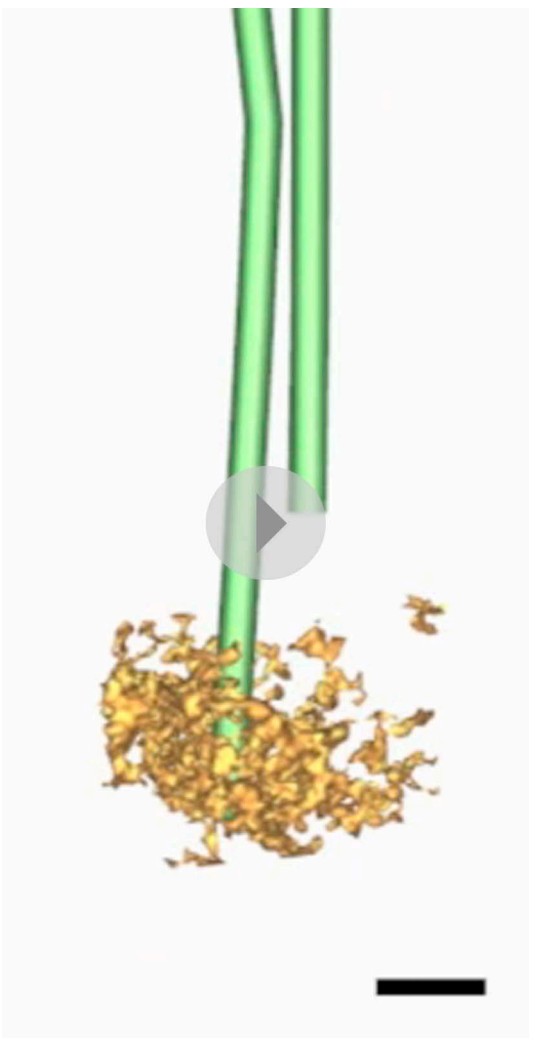

**Video 11**. The basal plate in a new *T. brucei* flagellum (related to *Figure 5*). The basal plate has been modeled by density thresholding and is displayed in brown. The CP is modeled as all MTs but also show their capped minus ends. Scale bar = 50 nm.

**Video 12**. The basal plate of an old *T. brucei* flagellum (related to *Figure 5*). The basal plate has been modeled by density thresholding and is displayed in brown. The CP is modeled and show one capped and one open minus end. Scale bar = 50 nm.

shed as a preparation for mitosis, so the cell is completely dependent on the reappearance of the two new flagella for its motility, making the rapid establishment of function important.

There are numerous further differences between the flagella of these two organisms: (1) The beat form of the flagella (*Schmidt and Eckert, 1976*; *Heddergott et al., 2012*). (2) The presence of a PFR in *T. brucei* (*Vickerman, 1962*; *Bastin et al., 1998*; *Portman and Gull, 2010*; *Höög et al., 2012*). (3) A rotational CP in *C. reinhardtii*; (*Mitchell, 2003*) vs a stationary CP *T. brucei* (*Gadelha et al., 2006*). Based on the findings presented in this paper, we can now add to this list a difference in the pathways for establishing axoneme organization during MT elongation, and maturation of the proximal region during the cell cycle.

It is important to note that we did not use deflagellation to create a situation of regenerating flagella in *C. reinhardtii*, as this regrowth might be different from the natural situation occurring when flagella regrow after mitosis. Pre-mitotic reabsorption of the flagellum probably allows for storage of the flagellar protein pool, a pool that would have been lost in the event of ionic shock deflagellation. Furthermore, in deflagellation before mitosis the basal plate is lost in a final event of

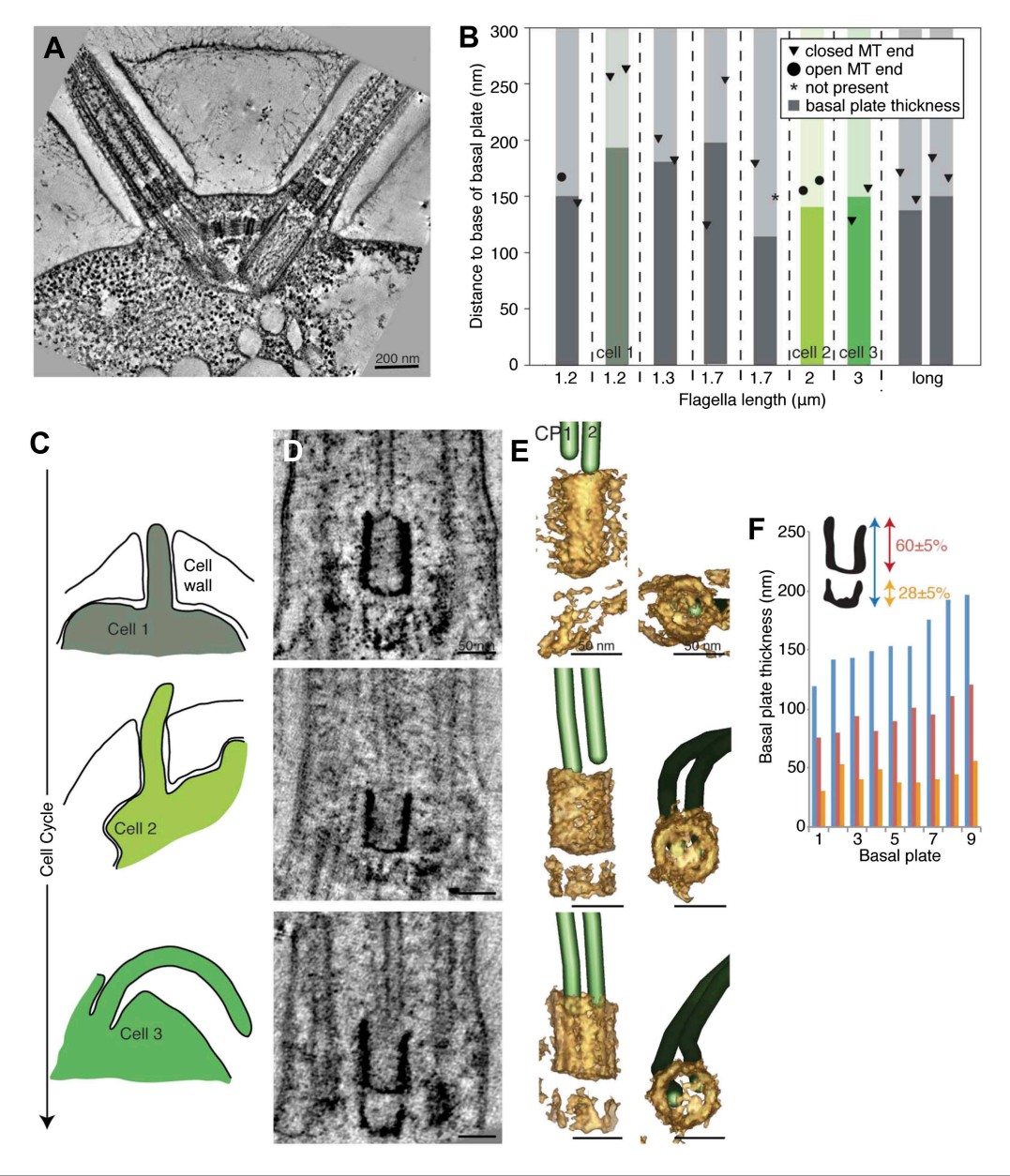

Figure 6. The transition zone in *C. reinhardtii* is structurally uniform during the cell cycle. (**A**) A 20-nm thick tomographic slice showing the two flagella extending from the *C. reinhardtii* cell. (**B**) The exact thickness of the transition zone central tube structure and the locations and structure of the CP minus ends within/around them. Note that the thickness of the central tube does not correlate with the flagellar length. This graph includes both flagella that we know are growing, and flagella of unknown dynamic state. The three colored columns represent the measurements of central tubes found in cell 1, 2 and 3 in **C–E**. (**C**) The cartoons to the left show outlines of the cells visualized in 10-nm thick tomographic slices of transition zone (**D**) and (**E**) where their 3D models show that the central tube structure consists of two baskets. The top basket is almost complete and the bottom one is partial. (**F**) The nine central tubes are here arranged by size, and the contribution of the upper and lower basket to the thickness of the structure is displayed. Note that the proportions within the structure remain similar through out, with the upper basket contributing 60 ± 5%, and the lower basket 28 ± 5% of the total central tube thickness.

shedding (*Parker et al., 2010*), whereas the stress induced deflagellation triggers a severing event distal to the basal plate, which can then form the transition zone for the next flagellum (*Rosenbaum et al., 1969*).

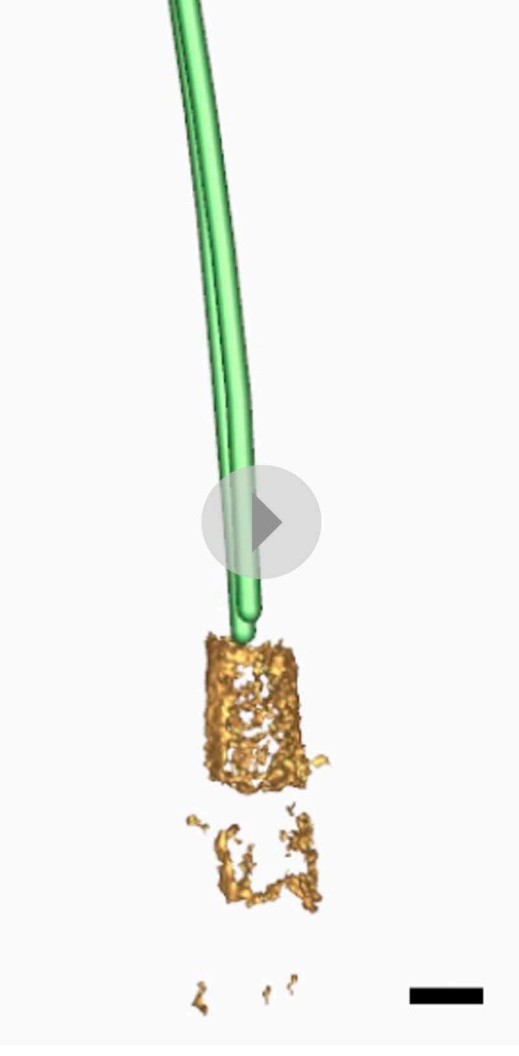

**Video 13**. The central cylinder in the *C. reinhardtii* transition zone (related to **Figure 6**). Central cylinder is modeled in brown, the CP in green. Both minus ends are capped. Scale bar = 50 nm.

We also revealed that the flagellum tip structure might be different to what was previously described. Most notably, we found no evidence of a singlet region, which has been shown to be over 2 µm long in *Tetrahymena sp* (**Sale and Satir, 1976**) and ~1 µm in *C. reinhardtii*. We also found no obvious CP cap, not disproving its existence (we did see evidence for filamentous proteins extending into the CP lumen), but showing that in well preserved cells it does not appear as previously described (**Dentler and Rosenbaum, 1977**; **Dentler, 1980**; **Fisch and Dupuis-Williams, 2012**). Even though we found the CPs to extend further than the dMTs, it was only within ~50 nm, in contrast to the published flagellar tip arrangement in *C. reinhardtii* (of unknown dynamic state), that shows the central pair protruding ~400 nm beyond the doublet microtubules (**Ringo, 1967**).

These differences in assembly order, speed, function, and structure, in two species both with the conventional 9+2 motile flagella structure, pose the question if the internal environments of flagella of different species are more different than presently assumed. Ciliogenesis will most likely involve a whole subset of flagellar proteins, with functions that are distinct from the IFT of axonemal proteins to the assembling tip. Flagellar growth rates after ionic shock in *C. reinhardtii* are length dependent, with shorter flagella growing faster than longer ones. Together with a steady disassembly rate, this forms the basis of the balance-point-model of flagellar length regulation (**Marshall et al., 2005**). The faster growth of short *C. reinhardtii* was then showed to depend on longer IFT trans in the short flagellum (**Engel et al., 2009**). This would fit well with our observations that the modes of assembly in the short and long post-mitotic *C. reinhardtii* tips do not differ beyond the point of central pair extension. Interestingly, we did detect differences in short vs long growing flagellar tips in *T. brucei*, an organism that most likely has linear flagellum growth. The absence of several axonemal components at the long growing tip might indicate that IFT is rate limiting in this growth. One would then predict that flagellar length is independent of size of IFT trains in these cells. However, it has recently been shown that the IFT traffic in *T. brucei* also differs considerably from that of *C. reinhardtii* (**Buisson et al., 2013**).

Thus, to further understand the normal structure and function of flagella, and the pathology of various ciliopathies, it is crucial to further understand ciliogenesis. With this paper we have revealed two modes of flagella growth, an area of flagellum biology that still remained mostly in the dark, and complementing our extensive knowledge on IFT of building material to the site of flagellar growth.

## Materials and methods

### Preparation *T. brucei* for electron tomography

Procyclic *T. brucei* strain 427 (high pressure freezing; HPF) or 29–13 (chemical fixation) were grown in SDM-79 media supplemented with 10% fetal bovine serum (for chemical fixation; PAA Laboratories Ltd, UK) and 20% fetal bovine serum (for HPF) (**Brun and Schönenberger, 1979**). Cell growth was

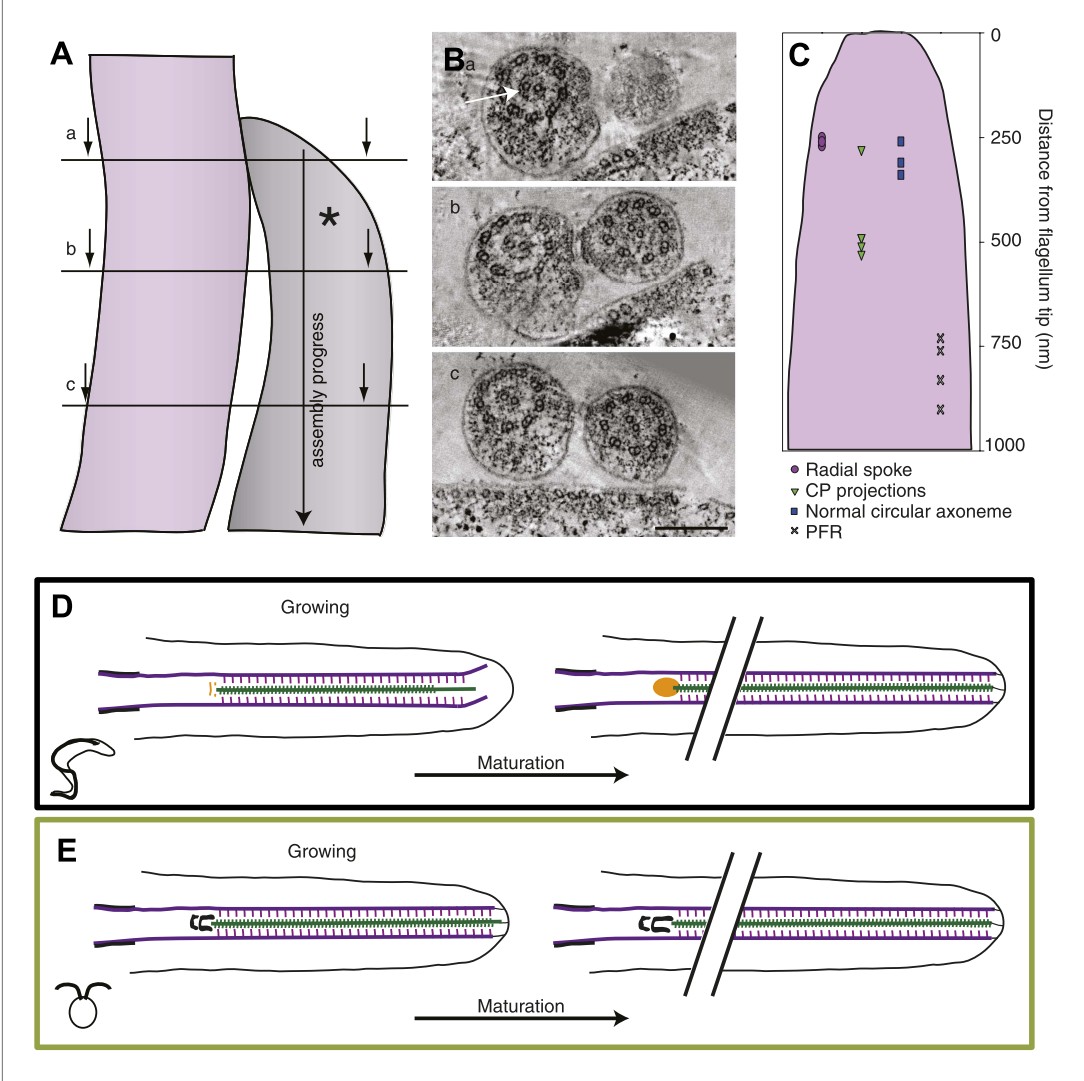

**Figure 7**. The axoneme is a timeline of flagellar assembly. (**A**) Because the new flagellum (*) grows at its distal tip, the axoneme is older closer to the basal body. This spatial assembly progress is used to reveal the order of structural additions to the growing axoneme. (**B**) Cross-sectional views from a tomogram of *T. brucei* flagella (old flagellum to the left and the growing new flagellum to the right). In (**A**) the slice is taken close to the tip of the growing flagellum. Only a few doublet microtubules have extended to here. (**B**) 100 nm in to the flagellum, dMTs and CPs are present but arranged in abnormal angles to each other. (**C**) The axoneme starts appearing normal ~250 nm into the axoneme. However, note the almost complete ring around the CPs in the old flagellum (arrow), most likely formed by central pair projections and/or heads of radial spokes. This ring is still not to be found in this most proximal tomographic slice taken. (**C**) Measurements of axonemal component incooperation was done in a multitude of cross-sectional and longitudinal flagella tip tomography reconstructions. (**D**) *T. brucei* flagellum grows in a disorganized manner probably due to a lack of associated axonemal proteins in the distal tip, this model is made to approximate where the components are added in comparison to the distal tip. As the flagellum matures (right), the axoneme is organized all the way to the tip and the basal plate has altered structure from the two rings to an electron-dense cloud. Electron-dense structures are visible from the lumen of the CP and A-tubule extending towards the membrane. (**E**) In *C. reinhardtii*, growth is organized with the CP protruding slightly at the tip of the axoneme. As the flagellum matures, the transition zone central tube extends but no morphological changes were seen.

monitored by using a CASY DT cell counter (Sedna Scientific, UK), and cultures were diluted on a daily basis to maintain a density between $5 \times 10^5$ and $1 \times 10^7$ cells per milliliter.

Cells were prepared for electron tomography by high pressure freezing or chemical fixation followed by epon embedding as described previously (*Höög et al., 2010*).

## Preparation of *T. brucei* for cryo-electron tomography

Unperturbed procyclic *T. brucei* strain 427 was plunge frozen and imaged intact as in (*Höög et al., 2012*).

## Preparation of *C. reinhardtii* for electron tomography

Wild-type *Chlamydomonas reinhardtii* strain 137C mt+ was grown at room temperature in liquid culture (Sagar and Granick medium) using a 10/14 dark/light cycle. After the cultures were shifted to the dark cycle, the cells were prepared for electron microscopy by HPF followed by freeze substitution as essentially described in *O'Toole et al. (2003, 2007)*. Briefly, the liquid culture was spun at 500×*g* for 5 min and the loose pellet was resuspended in a medium containing 150 mM mannitol for 1 hr. The samples were then spun at 500×*g*, the supernatant decanted and the loose pellet frozen using a BAL-TEC HPM-010 high-pressure freezer. The frozen samples were freeze substituted in 1% $OsO_4$ and 0.1% uranyl acetate in acetone for 3 days then embedded in epon/araldite resin.

## Serial sectioning and on-section staining

Semi-thick (300–400 nm) serial sections of the samples were cut using an ultracut UCT ultramicrotome (Leica Microsystems Ltd, UK). The sections were flattened by chloroform gas exposure whilst floating on the water surface. Ribbons of serial sections were put centered on 2 × 1 mm copper palladium slot grids (Agar Scientific Ltd, UK).

The sections were stained for 5 min on 2% uranyl acetate followed by 30 s Reynold's lead citrate. 15-nm colloidal gold particles were applied to both sides of the grid, to be used for image alignment.

## Electron tomography and image analysis

Tilt series of serial sections from flagella distal tips were acquired using the serialEM software (*Mastronarde, 2005*) operating a Tecnai TF30 300 kV IVEM microscope (FEI Co., The Netherlands). Images were collected about two orthogonal axes in 1° increments (±60°) using a Gatan CCD camera (pixel size 0.76–1.3 nm). Tomographic reconstructions were calculated, the two axis combined, serial sections stitched together and models were created using the IMOD software package (*Kremer et al., 1996*; *Mastronarde, 1997*).

Note that not all flagella tips are complete within the reconstructed volume in one tomogram. Many are reconstructed over serial sections. Measurements were only done within the same tomogram to ensure no error was introduced in the joining process. All flagella visualized were followed through the serial sections, taking lower magnification images of the entire cell to be able to reconstruct the flagellum length. Microtubule ends were classified as in *Höög et al. (2007, 2011, 2013)*.

## Acknowledgements

We thank Anthony Hyman, Mark Leaver, Per Widlund for critically reading the manuscript and Anthony Hyman for being a gracious host to JLH during the time of writing.

## Additional information

### Funding

| Funder | Grant reference number | Author |
|---|---|---|
| Wellcome Trust | Sir Henry Wellcome Postdoctoral Grant | Johanna L Höög |
| National Institutes of Health | 8P41GM103431-42 | Andreas Hoenger |
| EMBO | Long term postdoctoral fellowship | Johanna L Höög |
| Human Frontier Science Program | | Keith Gull |
| EP Abraham Trust | | Keith Gull |
| Henry Goodger Sholarship | | Sylvain Lacomble |
| Wellcome Trust | | Keith Gull |

The funders had no role in study design, data collection and interpretation, or the decision to submit the work for publication.

## Author contributions

JLH, Conception and design, Acquisition of data, Analysis and interpretation of data, Drafting or revising the article; SL, Acquisition of data, Analysis and interpretation of data, Drafting or revising the article; ETO'T, Conception and design, Acquisition of data, Drafting or revising the article; AH, Drafting or revising the article, Contributed unpublished essential data or reagents; JRM, KG, Conception and design, Analysis and interpretation of data, Drafting or revising the article

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
