## [Decision Letter]

Thank you for sending your work entitled “Mechanisms of flagellar assembly in *Chlamydomonas reinhardtii* and *Trypanosoma brucei*” for consideration at *eLife*. Your article has been favorably evaluated by a Senior editor and 3 reviewers, one of whom is a member of our Board of Reviewing Editors.

The Reviewing editor and the other reviewers discussed their comments before we reached this decision, and the Reviewing editor has assembled the following comments to help you prepare a revised submission.

All the reviewers found it an intriguing study. We like the idea that there are different modes of assembly, and the potential link to the structures seen in PCD patients. We would therefore like to publish your paper, as long as you address the concerns below.

The main concern for the reviewers is that you are over-stating your case. It may be that you are seeing different mechanisms, but you could easily be looking at different kinetics. We think that you need to discuss other ways of interpreting your data. We think you should call them different modes of assembly, rather than different mechanisms. You should comment on how the delayed incorporation of structural proteins that cross-link (and thus likely stabilize) the axonemal MTs may result in a less stable tip, which would in turn affect the kinetics of flagellar assembly (and the Balance-Point model is a good way to think about this). The reviewers also felt that given that you only have 5 tomograms per condition, the data set is a bit limited, and that you should state this caveat somewhere. When you resubmit, please answer these issues explicitly.

The reviewers had some discussions on whether you were always looking at growing flagella. We realised that because they are always within the mother wall, this means that they must be growing, but you should make this point explicit in the text.

You should also comment on why you did not use regeneration after pH shock, as this is a standard way to look at growth.

---

## [Author Response]

*The main concern for the reviewers is that you are over-stating your case. It may be that you are seeing different mechanisms, but you could easily be looking at different kinetics. We think that you need to discuss other ways of interpreting your data. We think you should call them different modes of assembly, rather than different mechanisms. You should comment on how the delayed incorporation of structural proteins that cross-link (and thus likely stabilize) the axonemal MTs may result in a less stable tip, which would in turn affect the kinetics of flagellar assembly (and the Balance-Point model is a good way to think about this)*.

We have changed the title to “Modes of assembly…” and we added the discussion around the Balance-Point model.

*The reviewers also felt that given that you only have 5 tomograms per condition, the data set is a bit limited, and that you should state this caveat somewhere*.

Throughout the paper, we clearly state our sample size (e.g., Table 2). Moreover, we might have had an average of 5 flagellar tips per condition, but many measurements were performed on dMTs and other structures within the axoneme that provided much better sample sizes. We therefore think that mentioning such a caveat as a general issue would misrepresent the data in the paper. However, we added this caveat to the end of the Results section.

*The reviewers had some discussions on whether you were always looking at growing flagella. We realised that because they are always within the mother wall, this means that they must be growing, but you should make this point explicit in the text*.

We refer to the first paragraph of the Results section: “*C. reinhardtii* has two flagella that are reabsorbed down to their transition zones, which are then expelled prior to mitosis (51; 46). After mitosis, the small daughter cells remain within the wall of the mother cell where they regrow their flagella; a cell stage we easily identified in the electron microscope.”

*You should also comment on why you did not use regeneration after pH shock, as this is a standard way to look at growth*.

In short, we did our experiment in a naturally occurring situation (regrowth after mitosis) that can be compared with the one found in *T. brucei*. Deflagellation via pH shock is a stress reaction and differs from natural flagellar reabsorbtion. For example, the mitotic cells also expel the transition zone (Parker et al*.*, 2010), whereas deflagellation via ionic shock leaves this structure behind (51).